# Developing a Stacked Ensemble-Based Classification Scheme to Predict Second Primary Cancers in Head and Neck Cancer Survivors

**DOI:** 10.3390/ijerph182312499

**Published:** 2021-11-27

**Authors:** Chi-Chang Chang, Tse-Hung Huang, Pei-Wei Shueng, Ssu-Han Chen, Chun-Chia Chen, Chi-Jie Lu, Yi-Ju Tseng

**Affiliations:** 1School of Medical Informatics, Chung Shan Medical University & IT Office, Chung Shan Medical University Hospital, Taichung 40201, Taiwan; threec@csmu.edu.tw; 2Department of Information Management, Ming Chuan University, Taoyuan 33300, Taiwan; 3Department of Traditional Chinese Medicine, Chang Gung Memorial Hospital, Keelung 20401, Taiwan; huangtsehung@gmail.com; 4School of Traditional Chinese Medicine, Chang Gung University, Taoyuan 33300, Taiwan; 5School of Nursing, National Taipei University of Nursing and Health Sciences, Taipei 11200, Taiwan; 6Graduate Institute of Health Industry Technology, Chang Gung University of Science and Technology, Taoyuan 33300, Taiwan; 7Department of Radiology, Division of Radiation Oncology, Far Eastern Memorial Hospital, New Taipei 22060, Taiwan; shuengsir@gmail.com; 8Faculty of Medicine, School of Medicine, National Yang Ming Chiao Tung University, Taipei 22060, Taiwan; 9Department of Industrial Engineering and Management, Ming Chi University of Technology, New Taipei 24330, Taiwan; 10Center for Artificial Intelligence & Data Science, Ming Chi University of Technology, New Taipei 24330, Taiwan; 11Institute of Medicine, Chung Shan Medical University, Taichung 40201, Taiwan; 12Department of Surgery, Division of Plastic Surgery, Chung Shan Medical University Hospital, Taichung 40201, Taiwan; 13Graduate Institute of Business Administration, Fu Jen Catholic University, New Taipei 242062, Taiwan; chijie.lu@gmail.com; 14Artificial Intelligence Development Center, Fu Jen Catholic University, New Taipei 242062, Taiwan; 15Department of Information Management, Fu Jen Catholic University, New Taipei 242062, Taiwan; 16Department of Information Management, National Central University, Taoyuan 32031, Taiwan; yjtseng.info@gmail.com

**Keywords:** head and neck cancer, stacked ensemble-based classification scheme, risk prediction, second primary cancers

## Abstract

Despite a considerable expansion in the present therapeutic repertoire for other malignancy managements, mortality from head and neck cancer (HNC) has not significantly improved in recent decades. Moreover, the second primary cancer (SPC) diagnoses increased in patients with HNC, but studies providing evidence to support SPCs prediction in HNC are lacking. Several base classifiers are integrated forming an ensemble meta-classifier using a stacked ensemble method to predict SPCs and find out relevant risk features in patients with HNC. The balanced accuracy and area under the curve (AUC) are over 0.761 and 0.847, with an approximately 2% and 3% increase, respectively, compared to the best individual base classifier. Our study found the top six ensemble risk features, such as body mass index, primary site of HNC, clinical nodal (N) status, primary site surgical margins, sex, and pathologic nodal (N) status. This will help clinicians screen HNC survivors before SPCs occur.

## 1. Introduction

Squamous cell carcinoma of the head and neck is a worldwide cancer, affecting a lot of people and causing deaths. Various genetic and environmental factors are related to head and neck cancers (HNC) [1], such as alcohol drinking, tobacco smoking, betel nuts chewing, and dietary factors [2,3,4]. However, little literature has mentioned prediction modules of second primary cancers (SPCs) in patients with HNC [5]. Therefore, understanding the risk factors driving HNC in patients with primary cancer affects the SPC diagnoses aimed at the highest risk. The Taiwan Cancer Registry database recorded 14 variables as clinical prognostic factors of HNC as follows: (1) age at diagnosis, (2) sex/gender, (3) primary site, (4) clinical stage group, (5) pathologic stage group, (6) combined stage group, (7) primary site surgical margins involvements, (8) lymph node size, (9) date of first surgical procedure, (10) radiotherapy and surgery sequence, (11) locoregional and systemic therapy sequence, (12) clinical target volume level (CTV_L: cGy) dosage, (13) lymph nodes in the neck into level I–III, and (14) body mass index (BMI). This study hypothesized a better-stacked ensemble method than individual classifiers to predict the possible risk factors of SPCs in HNC. Therefore, this study aimed to identify the most crucial risk factors from the 14 predictors listed for SPCs in HNC survivors.

## 2. Materials and Methods

### 2.1. Databases

A hospital-based cohort of 27,455 patients diagnosed with HNC was identified from the Cancer Registry data in Chang Gung Research Database, the largest multi-institutional database that includes de-identified electronic medical records from the Chang Gung Memorial Hospitals between 2004 and 2018. The Chang Gung Medical Foundation Institutional Review Board approved this study (IRB no. 201901386B0) and waived the requirement for patient consent.

Our study cohort randomly divided 27,455 cases, in which the training and testing datasets obtain 60% and 40%, respectively. Using 10-fold cross-validation, 60% of cases are used to train the base classifiers. The risk of SPCs for HNC in the 14 variables was compared using different base classifiers. The overall flowchart of the proposed method is shown in Figure 1.

### 2.2. A Stacked Ensemble-Based Classification Scheme

During the training process, 10 base classifiers are used. They are logistic regression (LGR), a classical linear method for binary classification; multivariate adaptive regression splines (MARS), a nonlinear regression model that is a combination of multiple regression models [6]; classification and regression tree (CART), a tree-based classification and regression method which uses recursive partitioning to split the data [7]; conditional inference trees (Ctree), a non-parametric class of decision trees and is also known as unbiased recursive partitioning [8]; C5.0, a program for inducing classification rules in the form of decision trees from a set of given examples [9]; evolutionary learning of globally optimal trees (EVtree), a procedure that implements an evolutionary algorithm for learning globally optimal classification and regression trees based on CART [10]; logistic model trees (LMT), a method that combines decision tree induction and logistic regression models [11]; random forest (RF), a supervised algorithm that uses an ensemble learning method consisting of a multitude of decision trees [12]; back-propagation neural network (BPNN), a multilayer, feed-forward neural network consisting of an input layer, a hidden layer, and an output layer [13]; and support vector machine (SVM), a discriminative classifier that is formally designed by a separative hyperplane [14]. These classifiers are modeled via the packages of “stats,” “earth,” “rpart,” “partykit,” “C50,” “evtree,” “RWeka,” “randomForest,” “nnet,” and “kernlab,” [15,16,17,18,19,20,21,22,23], respectively, under the R environment, version 3.5.1. The training procedure is conducted using the “caret” package [24]. The upsampling technique is applied to the minority class of patients with cancer suffering from SPCs to balance the number of cases of two classes. Moreover, 30 uniformly randomized hyper-parameter sets for each classifier are examined by 3 repeated 10-fold cross-validations to build a classifier whose evaluation metric, the area under the curve (AUC), is the highest.

The next step is to ensemble those tuned base classifiers, but not all of them. The goal is to build up a new classifier that utilizes strengths of each base classifier to improve the model performance. Empirical results suggest that a base classifier is a good candidate for an ensemble if predictions are fairly uncorrelated to others. Putting uncorrelated base classifiers together is encouraged because each captures a different dataset aspect. Thus, a classifier removal scheme is designed to sweep out classifiers whose predictions are highly correlated. The inter-classifier prediction correlations among base classifiers are calculated to form a correlation matrix. A base classifier pair is identified when the corresponding highest inter-classifier correlation within the upper triangular correlation matrix is larger than the threshold of 0.75 [25], as the correlation coefficient values between 0.7 and 1.0 indicate a strong positive linear relationship. The base classifier with higher average inter-classifier correlation to remaining base classifiers is removed. The above procedure is repeated until the highest inter-classifier correlation is less than the threshold.

After the base classifier removal, a meta-classifier is trained on the predictions of selected base classifiers using the “caretEnsemble” package [26]. The meta-classifier learns right or wrong base classifiers. An ultimate combination of base classifiers is produced, which improves the perdition performance even more. The importance of features for each base classifier is calculated. The importance of features for those base classifiers are sorted and then averaged by the weight of the overall model in the ensembled classifier, i.e., using a weighted sum rank aggregation method. Finally, the important ensemble features were extracted by estimating the knee point that fits two lines using linear regression [27].

After the meta-classifier organization is the final testing stage. Processed features are fed into the ensemble classifier to yield the response of each testing case. Each response will be compared with its corresponding label and then form a confusion matrix. With the confusion matrix available, several important metrics such as accuracy, sensitivity, specificity, and AUC are calculated to evaluate the model performance.

### 2.3. Removal Processes of Classifiers

The HNC database with 27,455 cases was randomly divided into 60% and 40% for training and validation, respectively. Using 10-fold cross-validation, 60% of cases were used to train the base classifiers. The tuned classifier SVM was removed at the beginning because it caused missing values. Then inter-classifier correlations among base classifier predictions were calculated as shown in Figure 2a. Next, filter base classifiers based on a threshold level of 0.75 get rid of the base classifiers with high inter-classifier correlations. This study revealed the highest inter-classifier correlations of 0.85 coming from LGR and C5.0. As shown in Figure 2b, the C5.0 was removed because of the higher inter-classifier correlation average than that of LGR. Based on Figure 2b, the pair of CART and LMT was continued because they had the highest inter-classifier correlations that were larger than the predefined threshold. As shown in Figure 2c, the CART was removed due to a larger inter-classifier correlation average. Therefore, C5.0 was removed, followed by CART during the removal processes. Finally, an LGR-weighted meta-classifier was used to ensemble collections of seven selected base classifiers.

The remaining 40% of cases were used to test the model performance. Accuracy, sensitivity, specificity, balanced accuracy, and AUC of testing datasets for each method are summarized in Table 1. The receiver operating characteristic curves of each classifier during the testing stage was also demonstrated in Figure 3. Additionally, RF receives the highest accuracy (0.842) and specificity (0.895) but very low sensitivity. The MARS received the highest sensitivity (0.754) but the lowest specificity. The balanced accuracy and AUC of base classifiers were no larger than >0.745 and 0.813. Contrarily, the performance of the ensemble meta-classifier on the testing dataset outperformed those of base classifiers. The balanced accuracy and AUC of the ensemble meta-classifier were as high as 0.761 and 0.847, which were approximately 2% and 3%, respectively, better than the best individual base classifier.

## 3. Results

The study data collection period from 2004 to 2018 revealed 27,455 datasets, including 3365 SPC cases. The stacked ensemble-based classification had higher balanced accuracy and AUC than other classifiers in this study. Additionally, the ensemble-based classifier with removal scheme increased both the balanced accuracy and AUC levels compared with that without the removal scheme (Table 1).

Moreover, by evaluating the prediction performance of each classifier, ranking the importance of the features provided useful information to identify risk factors for SPC of patients with HNC. All risk factors were listed according to their ranking by the final ensemble meta-classifier in Figure 4. The positive and negative effects of each feature were also identified using the information of LGR coefficient signs. Red bars mean negative relationship and blue bars show a positive relationship between the risk factors and SPCs. The results suggest that clinical N stage, lymph node size, lymph node metastasis to level I–III, combined stage, pathologic T status, dosage to CTV_L, and radiotherapy are negative risk features of SPCs of HNCs.

## 4. Discussion

Squamous cell HNC is a worldwide cancer, affecting a lot of people and causing deaths. Despite therapeutic developments in other malignancies, survival in HNC has not significantly improved in recent decades, especially in SPCs. Various genetic and environmental factors are related to HNC [1], such as alcohol drinking, tobacco smoking, betel nuts chewing, and dietary factors [2,3,4]. Recently, viral-related HNC became a major issue, such as human papillomavirus [28] and Epstein–Barr virus [29]. Moreover, patients with HNC who smoke at diagnosis have a significantly increased cancer death rate [30]. Additionally, older patients with HNC have a 1-year mortality rate of 42.3%, and malnutrition and immobility are independent negative predictors for worse survival [31]. However, the causes of SPCs in HNC survivors are rarely mentioned.

SPCs are a major cause of mortality in 13.2% of HNC survivors [32,33,34] due to field cancerization, denoting the entire aerodigestive epithelium exposure to carcinogenic insults, resulting in multiple premalignant and malignant lesions [35]. The SPC risk is approximately 2–4% per year, a rate of approximately 10–20% for overall lifetime risk [36].

With the effectiveness of early HNC screening and therapies in Taiwan, SPCs of HNC increased in our population and became a challenge for the healthcare systems. SPCs in patients with HNC occur most frequently in the head and neck region, lungs, and esophagus [37]. Bugter et al. showed that tobacco and alcohol consumption, comorbidity, and oral cavity subsite predicted the occurrence of SPC, and patients with a head and neck SPC more frequently received radiotherapy as locoregional therapy of their index tumor [38].

In 2015, Hollander Dd et al. demonstrated that patients with a higher BMI of ≥25 kg/m^2^ had increased overall survival and decreased disease-related mortality and recurrence rate compared to patients with underweight and normal weight [39]. Contrarily, in 2020, Peng Li et al. reported that clinical, T, and N stages were independent prognostic factors for patients with HNC [40]. BMI was associated with a higher probability of complications in HNC during therapies. However, no significant correlation was found in the overall, recurrence-free, and disease-specific survivals. In our study, BMI is the most important ensemble feature for SPCs in HNCs. However, the influence of BMI on SPC occurrence should be clarified in the future.

The clinical nodal stage, lymph node size, lymph node metastasis to level I–III, combined stage, and pathologic T status are all prognostic risk factors for primary HNC. Our study revealed that late clinical nodal stage, larger lymph node sizes, lymph node metastasis to level I–III, late combined stage, and late pathologic T stage all contributed to SPC occurrence in HNC survivors. Kuhlin B et al. demonstrated that primary tumor localization, age, sex, or TNM classification were not identified as significant indicators of the secondary carcinoma occurrence in 394 cases [41].

Gao et al. reported that radiotherapy carried a 68% excess risk of SPC development in the head and neck region in patients who survived >5 years after laryngeal cancer [42]. In our study, radiotherapy and the CTV_L (cGy) dosage played a negative feature.

In 2010, Chen et al. reported that men have a higher risk for a head and neck PC and SPC than women. The primary site, oral/pharynx, owned the highest frequency of second cancer (60%) development. Standardized incidence ratios were significantly higher for patients diagnosed while young, particularly 40 years old [43]. However, age at diagnosis is a positive feature in this study for head and neck SPC in HNC survivors.

## 5. Conclusions

A stacked ensemble method is firstly applied to predict the important features of SPCs for HNC survivors. The result reveals that this method can not only make a prediction but also ranked the important features of SPCs from several classifier combinations. Using an iterative base classifier removal scheme is necessary to achieve a better result during ensemble learning. In addition, this method also integrates the risk features from all individual base classifiers. This scheme could be also used in other diseases. For clinicians, the priority of these risk features was ranked. The importance of each risk feature can be revealed more easily.

Therefore, the finding suggested that clinical N stage, lymph node sizes, lymph nodes metastasis to level I–III, combined stage, pathologic T status, CTV_L dosage, and radiotherapy are negative risk features for SPCs of head and neck survivors, in order. These results will remind clinicians to pay attention to patients with such risk factors. We supposed nodal status and radiotherapy are two critical risk factors for SPCs in head and neck cancer survivors. We should pay more attention to such patients who owned these two factors. On the other hand, surgery, age at diagnosis, and pathological N status were positive risk features. As we know, surgical therapy is better to reduce the risk of SPCs in head and neck cancer survivors. Moreover, we should encourage head and cancer patients to receive surgical intervention to prevent SPCs if they can take the surgical risk.

## Figures and Tables

**Figure 1 ijerph-18-12499-f001:**
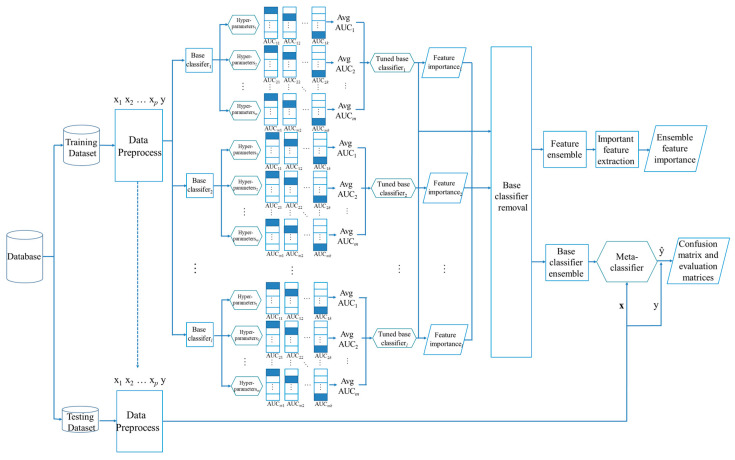
The overall flowchart of the proposed method, in which p, *l*, k, and m mean number of features, number of base classifiers, number of folds, and number of random values to try for each tuning hyper-parameter.

**Figure 2 ijerph-18-12499-f002:**
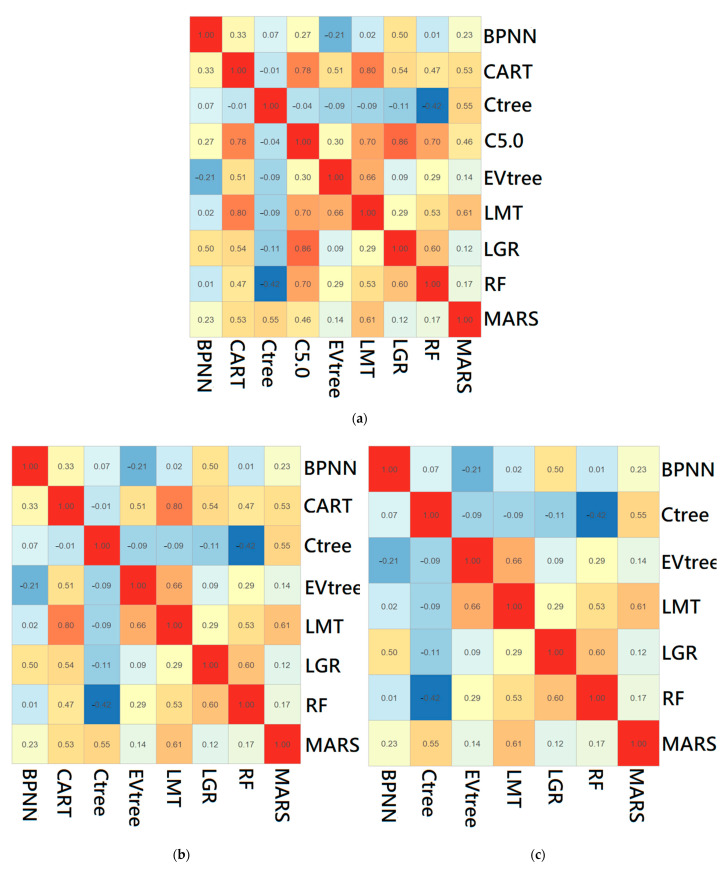
Inter-classifier correlations among base classifiers during training stage: (**a**) initial inter-classifier correlation matrix; (**b**) first iteration; (**c**) second iteration. Warm and cold color mean positive and negative inter-correlation, respectively. The darker the color, the stronger the degree.

**Figure 3 ijerph-18-12499-f003:**
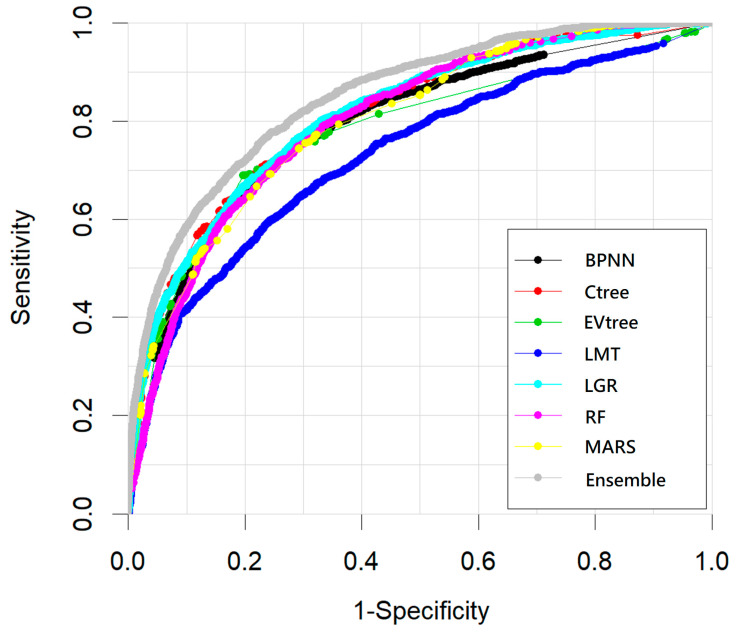
Receiver operating characteristics (ROCs) of all classifiers for testing dataset.

**Figure 4 ijerph-18-12499-f004:**
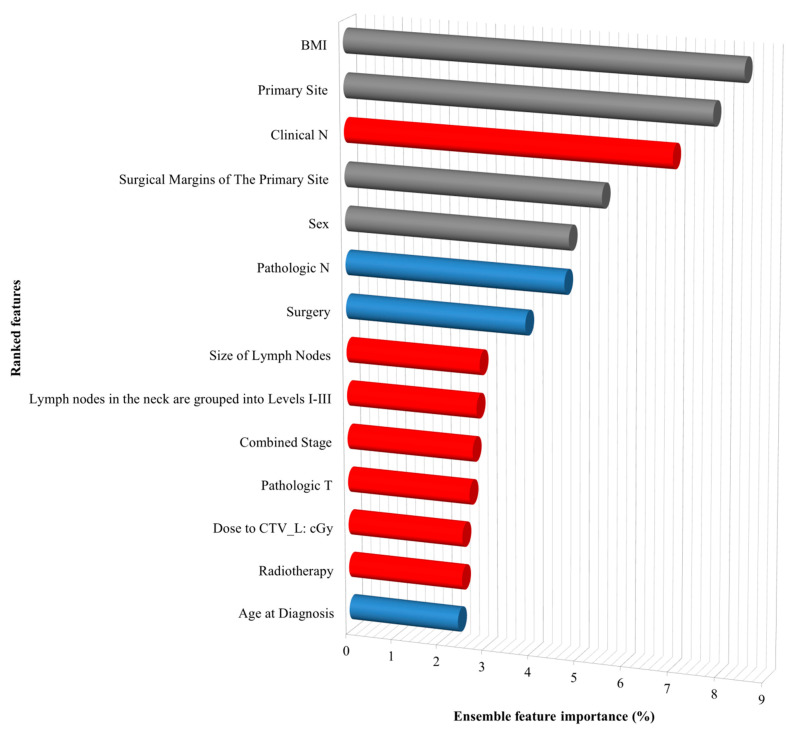
The 14 ensemble features importance for SPCs of head and neck cancers by the meta-classifier. “Direction” is based on the LGR analysis results and represents the direction of the correlation between features and the risk for SPCs. Red and blue bars mean negative and positive correlation, respectively. Grey bars mean categorical data. The most important feature ranked in the first place; on the contrary, the feature with the lowest importance is ranked as the last.

**Table 1 ijerph-18-12499-t001:** Model performance of base classifiers or meta-classifiers for testing dataset.

	Accuracy	Sensitivity	Specificity	Balanced Accuracy	AUC
LGR	0.759	0.699	0.767	0.733	0.813
MARS	0.703	**0.754**	0.696	0.725	0.800
Ctree	0.758	0.710	0.765	0.738	0.812
EVtree	0.786	0.689	0.800	0.745	0.777
LMT	0.818	0.452	0.869	0.661	0.729
RF	**0.842**	0.459	**0.895**	0.677	0.799
BPNN	0.746	0.715	0.750	0.733	0.791
Ensemble with base classifier removal scheme	0.778	0.738	0.783	**0.761**	**0.847**
Ensemble without base classifier removal scheme	0.782	0.705	0.792	0.749	0.836

Balanced accuracy meant the average value of sensitivity and specificity. AUC: Area Under Curve. Bold font indicates the best performance.

## Data Availability

Data are available from the Ethics Committee of the Chang Gung Memorial Hospital for researchers who meet the criteria for access to confidential data. Requests for the data may be sent to the Chang Gung Medical Foundation Institutional Review Board, Taoyuan City, Taiwan (e-mail: irb1@cgmh.org.tw).

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
