# Peer review of "Developing a Stacked Ensemble-Based Classification Scheme to Predict Second Primary Cancers in Head and Neck Cancer Survivors"

_ijerph, 2021, doi:10.3390/ijerph182312499_

Round 1
Reviewer 1 Report
The purpose of this study is to predict the most crucial risk factors from the 14 predictors listed for SPCs in Head and Neck Cancer Survivors. The clinical data set for the study were collected from 27,455 patients in a specialized hospital. The finding suggested that clinical N stage, lymph node sizes, lymph nodes metastasis to level I–III, combined stage, pathologic T status, CTV_L dosage, and radiotherapy are negative risk features for SPCs of head and neck survivors. The result reveals that this method use an iterative base classifier removal scheme is necessary to achieve a better result during ensemble learning. Overall speaking, the studied issue is important and practical. However, a major problem is that the authors do not explain the method clearly. Moreover, how these methods can be customized to the studied case should be clarified.
Major concerns:
1. The ten classification methods used in the proposed scheme were logistic regression (glm), multivariate adaptive regression splines (earth), classification and regression tree (rpart), conditional inference trees (ctree), C5.0, evolutionary learning of globally optimal trees (evtree), logistic model trees (LMT), random forest, back-propagation neural network (nnet), and support vector machine (svm). The reasons why these methods were used in this scheme should be discussed.
2. The abbreviation of each classification method should be carefully checked as some of that are unusual in literature. Based on this change, the figures 2 and 3, Table 1 and their corresponding description should be revised.
3. The citations of each R package are missing and the format should be checked.
4. The threshold in the proposed scheme should be discussed.
Author Response
Q1: The ten classification methods used in the proposed scheme were logistic regression (glm), multivariate adaptive regression splines (earth), classification and regression tree (rpart), conditional inference trees (ctree), C5.0, evolutionary learning of globally optimal trees (evtree), logistic model trees (LMT), random forest, back-propagation neural network (nnet), and support vector machine (svm). The reasons why these methods were used in this scheme should be discussed.
Response: Thanks for your suggestions. The short introduction of each method is included in the revised manuscript. The following description has been edited in the revised version.
“During the training process, ten base classifiers are used. They are logistic regression (LGR), a classical linear method for binary classification; multivariate adaptive regression splines (MARS), a nonlinear regression model that is a combination of multiple regression models [6]; classification and regression tree (CART), a tree-based classification and regression method which uses recursive partitioning to split the data [7]; conditional inference trees (Ctree), a non-parametric class of decision trees and is also known as unbiased recursive partitioning [8]; C5.0, a program for inducing classification rules in the form of decision trees from a set of given examples [9]; evolutionary learning of globally optimal trees (EVtree), a procedure that implements an evolutionary algorithm for learning globally optimal classification and regression trees based on CART [10]; logistic model trees (LMT), a method that combines decision tree induction and logistic regression models [11]; random forest (RF), a supervised algorithm that uses an ensemble learning method consisting of a multitude of decision trees [12]; back-propagation neural network (BPNN), is a multilayer, feed-forward neural network consisting of an input layer, hidden layer and an output layer [13]; support vector machine (SVM), a discriminative classifier that is formally designed by a separative hyperplane [14].”
Q2: The abbreviation of each classification method should be carefully checked as some of that are unusual in literature. Based on this change, the figures 2 and 3, Table 1 and their corresponding description should be revised.
Response: Thanks for your suggestions. The abbreviations of all classification methods are checked and Figures 2 and 3, Table 1 and corresponding description are modified in the revised manuscript.
Q3: The citations of each R package are missing and the format should be checked.
Response: Thanks for your suggestion. The citations are included in the references. Please see lines 100-102.
“These classifiers are modeled via the packages of “stats,” “earth,” “rpart,” “partykit,” “C5.0,” “evtree,” “RWeka,” “randomForest,” “nnet,” and “kernlab” [15-24] respectively under the R environment, version 3.5.1.”
Q4: The threshold in the proposed scheme should be discussed.
Response: Thanks for your comment. A base classifier pair is identified when the corresponding highest inter-classifier correlation within the correlation matrix upper triangular is larger than the threshold 0.75, as the correlation coefficient values between 0.7 and 1.0 indicate a strong positive linear relationship. The above description is included in the revised manuscript. Please see lines 116-119.

Reviewer 2 Report
interesting study and suggestion for follow our patients but how it use in everyday work
We know very well risk factor for developing SPC generally speaking, but at the moment is not possible to estimate risk factors for every patients and how apply it and have potentianal benefit.
Author Response
Q1: interesting study and suggestion for follow our patients but how it uses in everyday work?
Response: Thanks for your comments. In this study, we identified that clinical N stage, lymph node sizes, lymph nodes metastasis to level I–III, combined stage, pathologic T status, CTV_L dosage, and radiotherapy are negative risk features for SPCs of head and neck survivors, in order. We supposed nodal status and radiotherapy are two critical risk factors for SPCs in head and neck cancer survivors. We should pay more attention to such patients who owned these two factors.
In other hand, surgery, age at diagnosis and pathological N status were positive risk features. As we knew, surgical therapy is better to reduce the risk of SPCs in head and neck cancer survivors. We should encourage head and cancer patients to receive surgical intervention to prevent SPCs if they can take the surgical risk. For the elder patients, they seem to be at a lower risk of SPCs. These results will remind clinicians to pay attention to patients with such risk factors. Please refer to the Conclusion Section of revised manuscript.
Q2: We know very well risk factor for developing SPC generally speaking, but at the moment is not possible to estimate risk factors for every patient and how apply it and have protentional benefit.
Response: Thanks for your comments. Indeed, we need more evidences to estimate risk factors of SPCs for these head and neck cancer survivors. In this study, at least, we established a reliable stacked ensemble-based classification scheme and identified risk features to predict SPCs in head and neck cancer survivors. This scheme could be also used in other diseases. For clinicians, the priority of these risk features was ranked. The importance of each risk features can be revealed more easily. Please refer to the Conclusion Section of revised manuscript.

Reviewer 3 Report
The figure legends can be more explanatory
Author Response
The figure legends can be more explanatory.
Response: Thanks for your comment. We have added legends for Figures 1, 2, and 4 in the revised manuscript in order to make them more explanatory. The following description has been edited in the revised version.
“Figure 1. The overall flowchart of the proposed method, in which p, l, k, and m meant number of features, number of base classifier, number of fold, and number of random values to try for each tuning hyper-parameter.”
“Figure 2. Inter-classifier correlations among base classifiers during training stage. (a) initial inter-classifier correlation matrix; (b) first iteration; (c) second iteration. Warm and cold color meant positive and negative inter-correlation. The darker the color, the stronger the degree”
“Figure 4. The 14 ensemble features importance for SPCs of head and neck cancers by the meta-classifier. “Direction” is based on the LGR analysis results, represents the direction of the correlation between features and the risk for SPCs. Red and blue bars meant negative and positive correlation. Grey bars meant categorical data. The most important feature ranked in the first place; on the contrary, and the feature with the lowest importance is ranked as the last.”
